# Recent Advances in Aerobic Photo-Oxidation over Small-Sized IB Metal Nanoparticles

**Yifei Zhang [1,2], Meng Wang [3,*]** and **Gao Li [2,*]**

[1] Institute of Catalysis for Energy and Environment, College of Chemistry and Chemical Engineering, Shenyang Normal University, Shenyang 110034, China; sszyf2017@126.com

[2] State Key Laboratory of Catalysis, Dalian Institute of Chemical Physics, Chinese Academy of Sciences, Dalian 116023, China

[3] Key Laboratory of Biofuels and Biochemical Engineering, SINOPEC Dalian Research Institute of Petroleum and Petro-Chemicals, Dalian 116045, China

\* Correspondence: wangmeng.fshy@sinopec.com (M.W.); gaoli@dicp.ac.cn (G.L.)

**Abstract:** Aerobic photo-oxidation is a kind of green catalytic process that give valuable chemicals because of its mild reaction conditions and high product selectivity. Recently, small-sized IB metal nanoparticles (NPs; e.g., Cu, Ag, and Au, sized 1–3 nm) upon the surface of titanium oxide show excellent photocatalytic performance. The introduction of IB metal NPs can enhance the separation of photo-generated holes/electrons during photo-oxidations. In this account, we summarize the recent progress of small-sized IB metal NPs catalyzed by aerobic photo-oxidations, including the conversion of methanol, ethanol, sulfide, and benzylamine. More importantly, the structure–property correlations at the atomic level are detailed and discussed, e.g., the insights into the activation of oxygen and the identification of catalytic active sites. Future investigations are needed to carry out and reveal the catalytic mechanisms and conversion pathways.

**Keywords:** IB metal; 1–3 nm; structure–property correlations; nanoclusters; photo-oxidation





## 1. Introduction

In recent decades, well-defined IB metal nanoclusters (NCs) have been documented as novel and promising model catalysts owing to their unique electronic configuration [1–5]. For example, Au/Ag NCs show intrinsically different catalytic behaviors in comparison with conventional gold or silver nanoparticles (Au/Ag NPs) [6–10], which results in superior catalytic properties in many organic reactions, such as selective oxidation and hydrogenation and carbon-carbon couplings. Furthermore, the metal NCs with a nanometric size (1–2 nm) exhibit a non-metallic property with energy quantization manifested in their HOMO–LUMO gap, which is opposite to the localized surface plasmon resonance (LSPR) of conventional metal NPs [11,12]. The electron of the Au/Ag NCs can be excited from HOMO to LUMO upon illumination, resulting in the generation of electron/hole pairs as the semiconductors. The generation and subsequent recombination of the electron/hole pairs using solar energy are of great interest for photocatalytic and photovoltaics applications. Therefore, the catalytic activity of Au/Ag NCs has been demonstrated in solar energy harvesting for power production and the photocatalytic degradation of pollutants [13,14].

$TiO_2$-based semiconductors have been well documented in photo-oxidations [15–17]. Recently, supported small-sized $CuO_x$ NPs (3–5 nm), with a high population of surface oxygen vacancies, have been applied in a serial of redox reactions [18–21]. Firstly, these CuO NPs can aid the minimizing recombination of photo-generated electrons/holes, thus improving the photocatalytic activity. Secondly, the supported $CuO_x$ NPs can offer a unique electronic structure and synergistic effects, promoting photocatalytic reactions that occur at the interface of the photocatalysts. Therefore, the nanocomposites comprised of CuO NPs and $TiO_2$ semiconductors should be a good candidate for photocatalytic oxidation reactions.

Herein, we introduce the recent progress of photocatalytic transformations for valuable chemical productions, including the aerobic photo-oxidation of methanol to methyl formate over small-sized CuO supported onto $TiO_2$ in the gas phase, and the photocatalytic conversions of ethanol, sulfide, and benzylamine over Au/Ag NC photocatalysts, mainly in our lab until May 2022. Moreover, the extrapolate reaction pathways and tentative catalytic mechanisms, by density functional theory (DFT) studies, are detailed and discussed based on the systems developed.

## 2. Methanol Conversion to Methyl Formate

Methyl formate is one of the important chemicals applied in synthesis intermediates, the solvent for cellulose, and fumigants and fungicides [22–26]. The anatase-$TiO_2$ was shown to be a poor photocatalyst [27,28]; thus, the CuO/anatase-$TiO_2$ system was developed [29]. The previous studies only focused on titanium dioxide with mixed crystals (e.g., P25) as the support, which is not feasible to study the mechanism and the interaction of $TiO_2$ supports and metal NPs. Note that the interfacial perimeter of the oxide supports and metal NPs is often deemed as the catalytically active sites during the metal NPs catalyzed reactions.

In past decades, the well-defined anatase-$TiO_2$ with different shapes (such as rods, sheets, spindles, tubes, etc.) has been prepared [30,31]. $TiO_2$ has specific exposing facets; for example, $TiO_2\{001\}$ and $TiO_2\{101\}$ surfaces are observed in the $TiO_2$-sheet sample (Figure 1a), which have been widely used as photocatalysts [32,33]. The CuO/$TiO_2$-sheet photo-catalyst was prepared via a reduction-deposition method and used for the methanol photo-oxidation to methyl formate by oxygen ($O_2$) [26]. The CuO particles of ~3.5 nm were found to be supported on the $TiO_2\{101\}$ facet of the $TiO_2$-sheet. The CuO species can greatly reduce the recombination of photo-generated electrons/holes under light irradiation. The CuO/$TiO_2$ sheet exhibited excellent performance under mild conditions for the catalytic photo-oxidation of methanol. The activity of the CuO/$TiO_2$-sheet was improved by increasing the reaction temperatures, and the methyl formate selectivity decreased simultaneously.

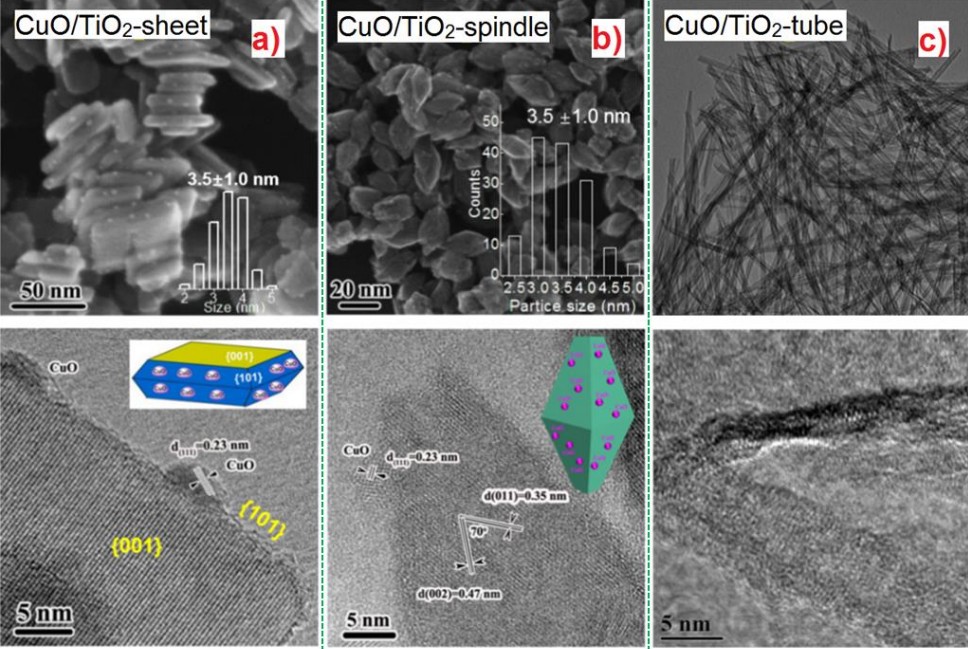

**Figure 1.** TEM images of the CuO/$TiO_2$ with different morphologies: (**a**) $TiO_2$-sheet, (**b**) $TiO_2$-spindles, and (**c**) $TiO_2$-tube. Reproduced with permission from Refs. [26,34,35]. Royal Society of Chemistry, 2019. Springer, 2020, and American Society of Chemistry, 2020.

The further investigation focused on the concentration of the $CuO_x$-loading when the conversion of a nearly 95% selectivity for methyl formate was larger than 85% could be reached with the $CuO_x/TiO_2$ photocatalyst; this was the best result in our reaction system (Figure 2). More $CO_2$ could be found when the concentration of $O_2$ increased to 0.75% and 1.0%, so 0.5% $O_2$ should be the most suitable choice by considering both the conversion and selectivity. At room temperature and optimal conditions, the reaction rate for methyl formate production can reach up to 10.8 mmol $g^{-1}$ $h^{-1}$, which is much higher than the used bare $CuO_x$ oxides or only a $TiO_2$-sheet. After a reaction for 20 h, the $CuO_x/TiO_2$-sheet nanocomposites still could show excellent catalytic activity. The synergistic effects and unique electronic structure were considered to be the main reasons for their superior catalytic activity.

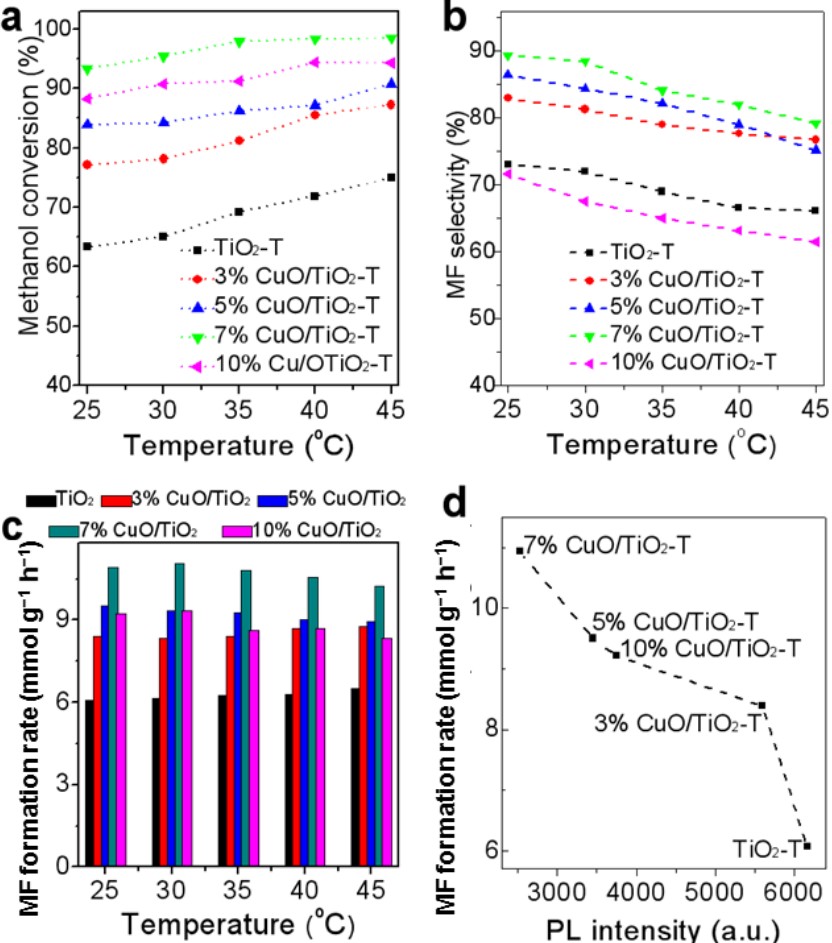

**Figure 2.** (**a**) $CH_3OH$ conversion, (**b**) MF selectivity, and (**c**) MF formation rate as a function of temperature over nanocomposites. (**d**) Relationship between PL intensity and MF formation rate. Reproduced with permission from Ref. [35]. American Society of Chemistry, 2020.

In the following investigation, Shi et al. supported the $CuO_x$ nanoclusters on the side of the $TiO_2\{101\}$ plane of a $TiO_2$-spindle (Figure 1b) [34]. It is necessary to mention that $CuO_x$ nanoclusters could not be anchored on the top of the $TiO_2$-spindle because there was not enough space. As a result, $CuO/TiO_2$ is a kind of excellent candidate for methanol oxidation; a conversion of >97% and 83% selectivity could be obtained under mild reaction conditions. Further investigation revealed that the presence of surface oxygen vacancy ($O_V$) species is an important factor for catalytic activity in the $CuO/TiO_2$-spindle catalyzed reactions [35]. Thus, the catalytic activity of small-sized CuO nanoparticles could be improved by tuning the excitons' recombination with $O_V$ generation.

Further, for the first time, in-situ attenuated total reflection infrared (ATR-IR) spectroscopic analysis reveals that the adsorbed methoxy (CH$_3$O*) was converted to an adsorbed formaldehyde (CHO*) species in the presence of oxygen in the methanol conversion (Figure 3a). The methyl formate was formed by a CHO* species that further reacted with a neighboring CH$_3$O* [34]. The reaction mechanism with the CuO/TiO$_2$ catalysts is shown in Figure 3b. With UV irradiation, electrons are formed at the TiO$_2$'s valence band and further transferred to its conduction band, coming out of the holes and electrons. Then, the accumulated electrons are transferred to the surface of CuO by a p-n heterojunction, which accelerates the separation speed of photo-generated electrons and holes, obviously, and the holes with positive charges could promote the formation of HCHO* formation by CH$_3$OH oxidation. A dissociation of oxygen occurred over the CuO particles with the aid of the generated electrons to refill the TiO$_2$ oxygen vacancies, which are the rate-determining steps during the methanol photo-oxidations [35].

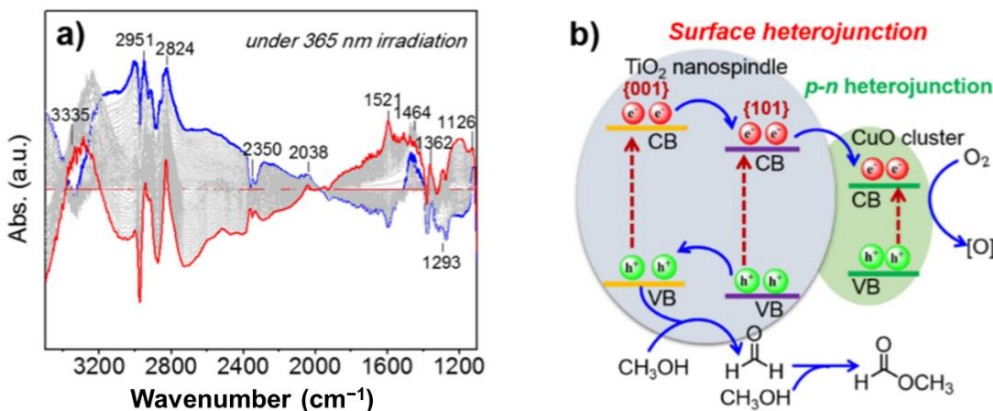

**Figure 3.** (**a**) ATR-IR monitoring of CuO/TiO$_2$ under 365 nm of light. (**b**) Proposed mechanism. Reproduced with permissions from Ref. [30]. Springer, 2019.

## 3. Ethanol Conversion

C9-C13 bio-fuels synthesized from cellulosic ethanol oxidation with cinnamaldehyde in a liquid have been well catalyzed by the nanogold particles supported by metal-oxides [35]. In a one-pot cascade cross aldol condensation reaction by K$_2$CO$_3$ as the cocatalyst, the Au/NiO catalyst could achieve a selectivity as high as 70% for C11-C13 hydrocarbon [36]. At the sites of NiO oxide's oxygen vacancies, ethanol was transformed into acetaldehyde (CH$_3$CHO*), which can be testified by EtOH-TPD and TGA analyses. Then, the reaction of cinnamaldehyde with CH$_3$CHO* happened at the interfacial perimeter of the Au/NiO composite through the cascade reactions. The whole catalytic process was investigated by in-situ infrared spectroscopy.

Atomically precise metal nanoclusters with certain structures also have been used in catalytic selective oxidations as a kind of well-defined model nanocatalyst [37–39]. Their electronic property could be tuned by mono-dopant into a metal particle at a specific position and thus change their catalytic activity. Qin et al. reported a method to dope a mono-Ag-atom at the central site of Au$_{13}$Ag$_{12}$(PPh$_3$)$_{10}$Cl$_8$ nanoclusters with a rod-shape and finally prepared a new kind of "pigeon-pair" cluster, which is called {[Au$_{13}$Ag$_{12}$(PPh$_3$)$_{10}$Cl$_8$]·[Au$_{12}$Ag$_{13}$(PPh$_3$)$_{10}$Cl$_8$]}$^{2+}$ [40,41]. The single-atom exchange between nanoclusters with the same structure resulted in an obvious disturbance to the electronic characters, which could lead to a difference in catalytic performance [42–44]. In order to investigate the influence of a single-atom exchange in the catalytic reaction, Au$_{13}$Ag$_{12}$ and Au$_{13}$Ag$_{12}$·Au$_{12}$Ag$_{13}$ clusters were both supported on TiO$_2$ and used in the photocatalytic conversion of ethanol [45]; the reaction proceeded under a UV irradiation at 30 °C. As can be seen from Figure 4, Au$_{13}$Ag$_{12}$·Au$_{12}$Ag$_{13}$ clusters achieved a higher conversion of ethanol, which is about 1.5-fold compared with the Au$_{13}$Ag$_{12}$ clusters (23%), and the selectivity of ethanal for the Au$_{13}$Ag$_{12}$·Au$_{12}$Ag$_{13}$ clusters (79%) is slightly higher

than the $Au_{13}Ag_{12}$ clusters (72%). Because the product distribution is very similar for the two different clusters, it could be concluded that the catalytic reaction mechanism and the conversion pathway over both catalysts should be the same. In brief, the single-atom exchange from Au to Ag in the same structure of the $M_{25}$ clusters leads to a significant difference in the catalytic activity caused by distinct electronic properties.

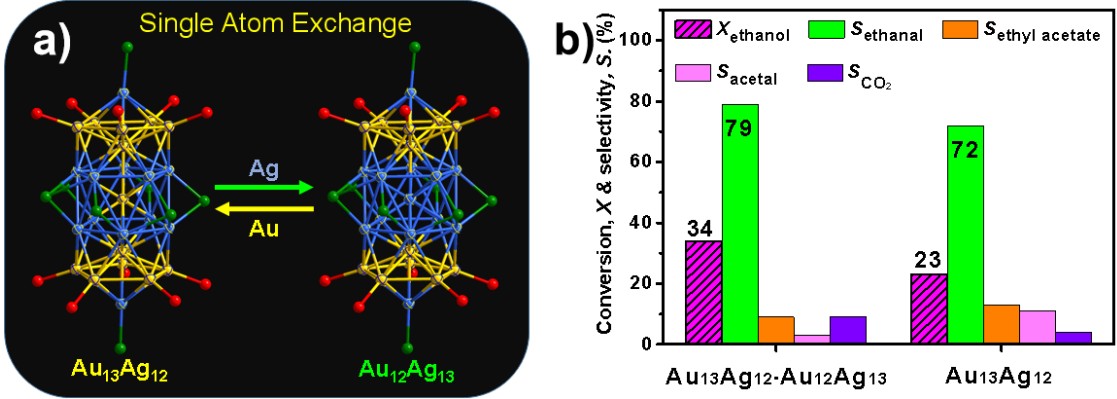

**Figure 4.** (**a**) Metal exchanging at the central site of the M25 clusters. (**b**) Photocatalytic performance in the ethanol conversion. Reproduced with permission from Ref. [46]. Springer, 2022.

## 4. Benzylamine to Imine

The catalytic selective oxidation of amines to corresponding imines is a kind of significant and important reaction due to the extensive applications in the industry of fine chemicals and pharmaceuticals [47]. Tada and coworkers reported a plasmonic gold photocatalyst which gave a 4.5% conversion of benzylamine to imine without a solvent [48]. The metal clusters show a good capacity to adsorb UV/VIS lights [49,50]. With the aim to expand the application of support for Au nano-cluster catalysts in this reaction system, we anchored $[Au_{25}(PPh_3)_{10}Cl_2(SR)_5]^{2+}$ rods with a HOMO-LUMO gap of ~1.84 eV ($>\Delta E_{act}$) on a P25 (mixed anatase and the rutile phase of $TiO_2$) to obtain $Au_{25}$/P25 nanocomposites with ligands intact on the cluster's surface and used it in the photo-oxidation of benzylamine under visible light ($\lambda \approx 455$ nm) in the presence of $O_2$ [51]. After 2 h of light irradiation, the conversion was 82% with the $Au_{25}$/P25, which is much higher than bare P25 (about 14%) in the same reaction conditions. After three times of cycle use, only a 10% decrease in activity happened, and the selectivity was still maintained at >99%.

Figure 5b shows a plausible catalytic reaction mechanism; under light irradiation, $Au_{25}$ rods could act as a narrow band gap semiconductor and lead to the separation of the holes and electrons. After being generated in the $Au_{25}$ clusters by photo-oxidation, electrons could be transferred into the conduction band of P25 and react with $O_2$ to form $O_2^-$. Because the LUMO energy of $Au_{25}$ is higher than that of the conduction band of P25 ($-0.63$ V to $-0.9$ V), the electrons could inject into the conduction band of P25 from $Au_{25}$. Firstly, on the surface of the nanocluster, $\alpha$-H in the benzyl amine radical cation can be plucked by a Au atom without an organic ligand. By inducing amines, the phosphine ligands could be partially removed from the Au atoms; this process has been proved by experiment and theoretical calculations [52,53]. As a result, a Au-H intermediate, along with a carbocation intermediate, would be formed. The H atom could be withdrawn from the Au-H and amine by the $O_2^-$ species to form $PhCH = NH$ and $H_2O_2$. The benzyalamine would be oxidized by $H_2O_2$ to produce water and benzaldimine, which would undergo a nucleophilic attack by benzylamine to form aminal lately. Finally, the aminal group eliminates ammonia to produce an imine product.

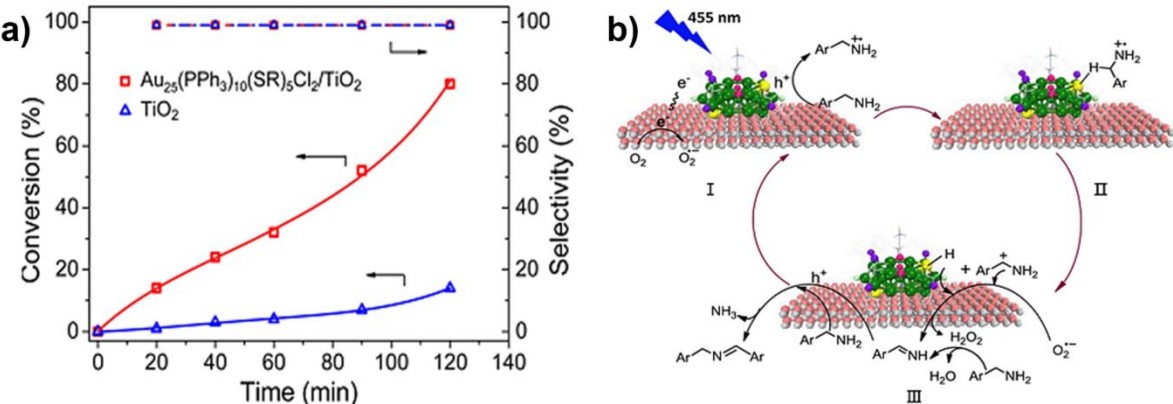

**Figure 5.** (**a**) Time-dependent catalytic activities of Au$_{25}$/P25 and P25 in the photo-oxidation of benzyl amine. (**b**) Mechanistic process of photo-oxidation. Reproduced with permission from Ref. [52]. Copyright 2017 American Chemical Society.

In a further experiment, a highly photostable novel alloy Au$_8$Ag$_3$(PPh$_3$)$_7$Cl$_3$ cluster (E$_g$~1.67 eV) was used in the photo-oxidation of benzyl amine to *N*-(phenylmethylene) ben-zenemethanamine under an LED light (λ > 455 nm) with the presence of O$_2$ at room temperature, Figure 6 [54]. The catalytic activity was also compared with its common monometallic counterpart, Au$_{11}$(PPh$_3$)$_7$Cl$_3$ (Eg~2.06 eV). Both the Au$_8$Ag$_3$/P25 and Au$_{11}$/P25 (0.5 wt.% cluster loading) were used without organic ligand removal. At the same reaction conditions, Au$_8$Ag$_3$/P25 had a 72.5% conversion compared with Au$_{11}$/P25, which only had a 37.8% conversion, and the selectivity for both was >99%. The turnover frequency for Au$_8$Ag$_3$/P25 was about two times higher than Au$_{11}$/P25 (1.51 s$^{-1}$); this could be attributed to the special electronic properties of the Ag$_8$Au$_3$ alloy cluster caused by the synergic effects from the Au and Ag atoms. A DFT investigation further revealed that Ag dopants in the C$_3$-axis delocalized the electrons of Au into the orbitals of the phosphorus atom and induced a disturbance in the electronic properties, which could affect the catalytic activity. It is worth noting that the reaction could not happen without light irradiation.

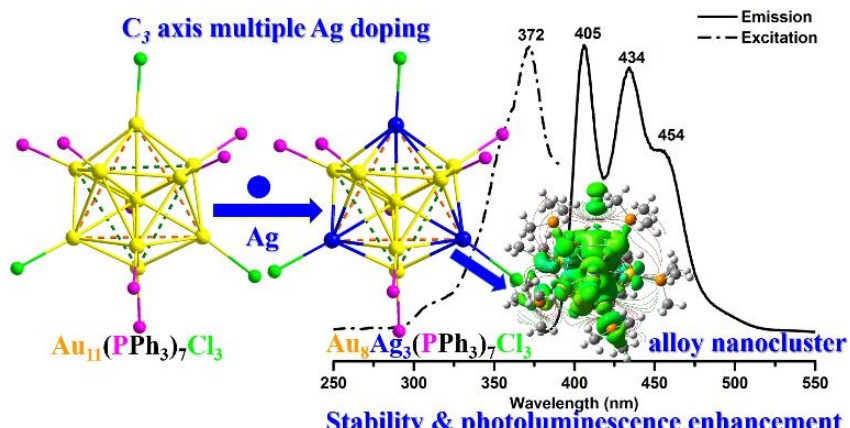

**Figure 6.** Fabrication of Au$_8$Ag$_3$(PPh$_3$)$_7$Cl$_3$ for the photocatalytic conversion of benzylamine to imine. Reprinted from Ref. [54]. Copyright 2019 Royal Society of Chemistry.

## 5. Sulfide to Sulfoxide

Singlet oxygen ($^1$O$_2$) is a kind of excited state of molecular oxygen ($^3$O$_2$ or triplet oxygen, i.e., a ground state). Because of its high reactivity, $^1$O$_2$ could take part in various kinds of chemical and biological reactions. However, according to the selection rule, the direct conversion between $^3$O$_2$ and $^1$O$_2$ spin states via photons' adsorption/emission is spin-forbidden [55]. In order to solve this issue, the photosensitizers (photo-absorbing

molecules) are usually used for the photo-mediated generation of $^1O_2$, which have to satisfy that the energy difference between the triplet and ground state of the photosensitizer ($\Delta E_t$) should be larger than the activation energy of the triplet oxygen ($^3O_2$ to $^1O_2$ with $\Delta E_{act} \approx 0.97$ eV). Interestingly, different from metal nanoparticles, the metal nanoclusters with atomically precise structures exhibit HOMO-LUMO gaps ($E_g$) and $Au_{25}(SR)_{18}$, $Au_{38}(SR)_{24}$, and $Au_{99}(SR)_{42}$ clusters have an $E_g$ of ~1.3, 0.9, and 0.71 eV, respectively [56]. So, Kawasaki et al. demonstrated the fact that the HOMO-LUMO gap (~1.3 eV > $\Delta E_{act}$) of $Au_{25}(SR)_{18}$ permits it to be used in the production of $^1O_2$ [57]. The formation of $^1O_2$ through photo excitation with a cluster of such type ultimately leads to the transformation of various organic compounds. Mechanistically, the photo-excited sensitizer passed on its excess energy to $^3O_2$ to generate $^1O_2$, and the photosensitizer regenerated in the ground state without any consumption. Additionally, some superoxide ($O_2^{\cdot -}$) and hydroxyl radicals ($^{\cdot}OH$) may also be formed at the same time via other photochemical reactions, such as an electron transfer process [58]. Next, these kinds of phenomena will be explained by demonstrating some of the key examples from our lab.

A novel structure called the $Au_{38}S_2(SAdm)_{20}$ cluster (size ~1.5 ± 0.3 nm) with a HOMO-LUMO gap of 1.57 eV (>$\Delta E_{act}$) was found to be much more efficient at generating $^1O_2$ under a laser light of 532 and 650 nm [59]. As can be seen from Figure 7, the reaction mechanism involves a Dexter-Type electron exchange coupling between the excited photo-sensitizer and the ground state $^3O_2$ molecules. Briefly, the $Au_{38}S_2(SAdm)_{20}$ cluster was stimulated from its ground state ($S_0$) to an excited state ($S_1$) by photo excitation, then converted to a slightly lower energy state ($T_1$) with spin flipping via an intersystem crossing (ISC). $Au_{38}S_2(SAdm)_{20}$ in the ground state and the much-required $^1O_2$ could be generated by an electron exchange between the triplet state ($T_1$) of the $Au_{38}S_2(SAdm)_{20}$ cluster and $^3O_2$.

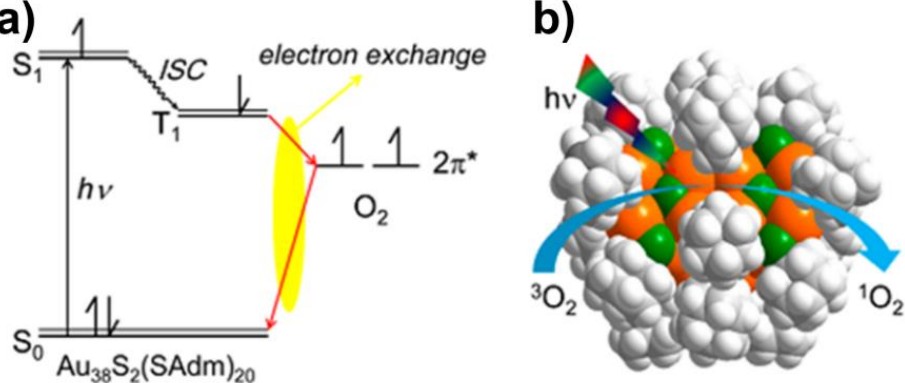

**Figure 7.** (**a**) Mechanism of the Dexter-Type electron exchange for $^1O_2$ generation. (**b**) Description of the generation process. Reprinted with permission from [59]. Copyright 2017 American Chemical Society.

Then, the $Au_{38}S_2(SAdm)_{20}$ cluster was further used for photocatalytic sulfoxidation reactions in an open quartz vessel with $O_2$ bubbling under light irradiation ($\lambda = 532$ nm). As earlier reported, gold nanoclusters could display excellent catalytic performance in the selective oxidation of sulphides to sulfoxides using a chemical oxidant, e.g., PhIO (without any light irradiation), and this reaction system results in over-oxidized impurities, such as sulfones [60]. Photocatalytic sulfoxidation with molecular oxygen as the oxidant is more economical and environmentally friendly.

It is worth noting that no catalytic activity could be observed in the absence of light; under light at $\lambda \approx 532$ nm, the catalytic activities of the $Au_{25}(SR)_{18}$ and $Au_{38}S_2(SAdm)_{20}$ clusters could achieve up to 18% and 57%, with 100% selectivity for methyl phenyl sulfoxide. After three reaction cycles, no loss of catalytic activity and selectivity for $Au_{38}S_2(SAdm)_{20}$ was observed. The high efficiency of the $^1O_2$ generation with $Au_{38}S_2(SAdm)_{20}$ prompted the further investigation of the photocatalytic activity of the $Au_{38}S_2(SAdm)_{20}$ nanoclusters used in a one-step selective chemical transformation of benzylamine to

*N*-(phenylmethylene)benzenemethanamine under an LED light ($\lambda \approx 455$ nm) in the atmosphere of $O_2$. In the same reactions, a 99% conversion of the substrate was reached in 0.5 h with >99% selectivity; this result is much better than supported Au nanoparticles (6–12 nm) [61]. As expected, the catalytic activity for the $Au_{38}(PET)_{24}$ clusters, with a HOMO-LUMO gap of ~0.9 eV (<$\Delta E_{act}$), decreased sharply (only by ~20% in the benzylamine conversion); it obviously indicates that the high catalytic conversion (99%) of $Au_{38}S_2(SAdm)_{20}$ is due to the formation of a singlet $^1O_2$ during the photocatalytic reaction. Importantly, part of the photocatalytic oxidation caused by the Au nanoclusters should also happen via the electron transfer process and the production of the benzyl amine cation.

Next, the $[Au_{13}(dppe)_5Cl_2]^{3+}$ ($Au_{13}$) NC with a propeller-like structure and a HOMO-LUMO gap of ~1.9 eV had a high $^1O_2$ quantum yield of 0.71 [62], which is considerably close to that of anthracene. Of note, the $Au_{13}$ NCs were intact and stable during the $^1O_2$ generation.

## 6. Conclusions, Challenges, and Future Perspective

In summary, high valuable chemical products synthesized by aerobic photocatalytic oxidation with supported IB metal nanoclusters have been fabricated in this study; some significant breakthroughs are summarized below:

(1) Four different types of photo-oxidations were achieved over the $TiO_2$-supported metal nanoclusters and nanoparticles presented in this paper, including the photo-oxidation of methanol to methyl formate, and ethanol conversion, a benzylamine to imine conversion, and sulfide to sulfoxide conversion, which give some new exploits in photocatalysis.

(2) Copper oxide nanoclusters supported onto the $TiO_2$ semiconductors with a different morphology of nanosheets, nanospindles, and nanotubes have been prepared to investigate the performance of photocatalytic oxidation reactions. The morphology (exposed facets) effects have been well studied.

(3) The active centre for photocatalytic aerobic oxidation was identified and created as an example. The $CuO_x/TiO_2\{101\}$ interface was considered to be a key point in the photo-oxidation of methanol.

(4) Single-atom-exchanging in the metal clusters has a big influence on the catalytic activity but cannot affect product selectivity. However, the detailed mechanism is still unable to reveal this.

(5) Via the characterization of ATR-IR spectra to dig out the possible reaction intermediates [63], we have clearly mapped out the whole conversion pathway and worked out the controversy in this photocatalysis.

However, there are still some following points for utmost consideration:

(1) More efforts must be put into the creation of novel photo/catalysts with specific active reaction sites for a high-efficient conversion [64–66], and the catalytic efficiency should be further enhanced by the design of the novel metal nanoclusters or nanoparticles on other light-absorbing nanomaterials (e.g., ZnO, g-$C_3N_4$, BiOX, Mxenes, etc.). The photocatalysis also needs to be developed using UV/VIS light, which is close to the real applications.

(2) In order to reveal the reaction mechanisms, DFT studies and in-situ characterizations should be used in this catalytic reaction system, and, in turn, it will be more helpful in obtaining new photocatalysts with high efficiency.

(3) The photo-oxidation reactions should be further coupled with a thermal and electro method to improve the catalytic activity, which needs to develop a new type of photocatalyst [67–72].

(4) Alloy metal nanoparticle catalysts still need to be further and wildly used in catalytic biomass conversions; as the electronic property can be well modified to tune the catalytic performance, it is a kind of excellent method for designing suitable catalysts.

**Author Contributions:** Investigation, Y.Z. and M.W.; resources, Y.Z. and M.W.; writing—original draft preparation, Y.Z., M.W. and G.L.; writing—review and editing, G.L.; supervision, G.L.; project administration, M.W. and G.L. All authors have read and agreed to the published version of the manuscript.

**Funding:** We gratefully acknowledge the financial support from the Innovation Foundation of Dalian Institute of Chemical Physics (DMTO201802).

**Institutional Review Board Statement:** The study was conducted in accordance with the Declaration of Helsinki and approved by the Institutional Review Board.

**Informed Consent Statement:** Not applicable.

**Conflicts of Interest:** The authors declare no conflict of interest.

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
