# Peer review of "Recent Advances in Aerobic Photo-Oxidation over Small-Sized IB Metal Nanoparticles"

_2673-7256, doi:10.3390/photochem2030037_

Round 1

Author Response

The author should mention, why anatase TiO2 is chosen as supporting material for CuO nanoparticles, particularly in the introduction section. Please extend your discussion of other photocatalytic materials and its relevance in introduction section.

A: Many thanks for the suggestion. We have added in the introduction section. And the related work is cited.

The necessity and novelty of constructing small-sized IB metal 13 nanoparticles (NPs, e.g. Cu, Ag, Au, size: 1-3 nm) upon the surface of titanium oxide need to be clarified in the introduction section. Please go through the related work to bimetallic nanoparticles.

A: The necessity and novelty have been clarified in the introduction section. And the related work is cited.

The resolution of the figures should be improved in the revised manuscript, Figure 3, 5, and 6.

A: Many thanks for the suggestion. It has been done.

Conclusion is too confined and it is suggested to write it in more detail.

A: Many thanks for the suggestion. We have rewritten the Conclusion in details.

Reviewer 2 Report

Green catalytic processes are of great interest nowadays. This paper has an important contribution summarizing some recent research of the small-sized IB metal nanoparticles catalyzed aerobic photo-oxidations reactions, highlighting the structure-property correlations and the identification of catalytic active sites.

However, this paper requires major revisions to be published in Photochem.

Attached are some comments for the authors.

Author Response

The difference (if any) between nanoparticles and nanoclusters is not very clear, perhaps a discussion should be introduced (for example, in Cap 2. Methanol conversion to methyl formate-where CuO/TiO2 is discussed);

A: Many thanks for the suggestion. The discussion of nanoparticles and nanoclusters is added.

Some Figures (for example, Figures 1, 2, 3) and results were also presented in another published review (Zhang, Y., Cao, C., & Li, G. (2022). Recent Progress in Green Conversion of Biomass Alcohol to Chemicals via Aerobic Oxidation. Biomass, 2(2), 103-115.);

A: Figures 1, 2, 3 are similar with the work on Biomass. But these Figures have been rechanged to distinguish them in Biomass.

-Some Figures and Tables that are presented in the text do not actually exist or are

misspelled;

A: Sorry, it is our mistake. We have revised them one by one in the revised manuscript.

Round 2

Reviewer 1 Report

The revisions are OK

Reviewer 2 Report

The manuscript has been corrected and improved. I think most of my comments have been addressed with necessary discussions or corrections. I recommend the acceptance of the manuscript.
